# Small Vessel Disease-Related Dementia: An Invalid Neurovascular Coupling?

**DOI:** 10.3390/ijms21031095

**Published:** 2020-02-07

**Authors:** Rita Moretti, Paola Caruso

**Affiliations:** Neurology Clinic, Department of Medical, Surgical and Health Sciences, University of Trieste, 34149 Trieste, Italy; paolacaruso83@gmail.com

**Keywords:** small vessel disease, vascular damage, endothelium, neurovascular coupling, inflammation, oxidative response, redox, brain’s autoregulation

## Abstract

The arteriosclerosis-dependent alteration of brain perfusion is one of the major determinants in small vessel disease, since small vessels have a pivotal role in the brain’s autoregulation. Nevertheless, as far as we know, endothelium distress can potentiate the flow dysregulation and lead to subcortical vascular dementia that is related to small vessel disease (SVD), also being defined as subcortical vascular dementia (sVAD), as well as microglia activation, chronic hypoxia and hypoperfusion, vessel-tone dysregulation, altered astrocytes, and pericytes functioning blood-brain barrier disruption. The molecular basis of this pathology remains controversial. The apparent consequence (or a first event, too) is the macroscopic alteration of the neurovascular coupling. Here, we examined the possible mechanisms that lead a healthy aging process towards subcortical dementia. We remarked that SVD and white matter abnormalities related to age could be accelerated and potentiated by different vascular risk factors. Vascular function changes can be heavily influenced by genetic and epigenetic factors, which are, to the best of our knowledge, mostly unknown. Metabolic demands, active neurovascular coupling, correct glymphatic process, and adequate oxidative and inflammatory responses could be bulwarks in defense of the correct aging process; their impairments lead to a potentially catastrophic and non-reversible condition.

## 1. Introduction

Cerebral small vessel disease (SVD) primarily distresses the small perforating arteries, being defined as vessels with less than 50 µm diameters, also defined as “all the vessels within the brain parenchyma plus the vessels with a diameter less than 500 µm in the leptomeningeal space” supplying the deep brain structures [1,2]. Nevertheless, general increased arterial stiffness is associated with an increased white matter lesion burden [3]. Therefore, while the microvasculature is the primary target of SVD, the contribution of larger arteries should not be immediately discounted. SVD is the most important and common cause of vascular dementia, leading to 45% of dementia, and it accounts for about 20–30% of all strokes worldwide, 25% of ischemic (or lacunar strokes). Moreover, it significantly increases the risk of future stroke [4]. Often, SVD lesions are clinically insidious and they act as “silent” lesions. Thus, SVD is a dynamic pathology, lesions progress over time, and the long-term outcome and impact on brain damage vary [5]. In sporadic cerebral SVD, sporadic aging and hypertension are listed as the most critical risk factors. However, different hereditary forms of cerebral SVD have also been described [6]. In the latter forms, several pathological changes to the vasculature in small arterioles (like vascular muscle dysfunction, lipohyalinosis, vascular remodeling, and the deposition of fibrotic material) have been identified. Venous structures are also affected [7]. These facts are shared in both forms, with time of onset beng the only difference.

Cerebral amyloid angiopathy (CAA) is a common form of cerebral SVD and it refers to the deposition of amyloid b-peptide (Ab) in the cerebral leptomeningeal and parenchymal arteries and arterioles walls. The incidence of CAA increases with age. More often, CAA is related to hemorrhagic stroke. Additionally, in this case cause, structural variations, such as concentric splitting, loss of smooth muscle cells, and fibrinoid necrosis, which may increase the propensity for vessel rupture and, thus, hemorrhage, have been seen [8,9].

## 2. Vascular Dementia and Small Vessel Disease-Related Dementia

The diagnosis of vascular dementia should be easy due to the temporal correlation between an acute vascular brain lesion and the onset of cognitive and behavioral problems. Nonetheless, consensus criteria for vascular cognitive impairment are still under debate, since 1983, when NINDS-AIREN Criteria had first been written [10,11], and the ICD-10 had been debated [12]. Subsequently, different validations have been proposed, and many criteria have been written, but the current clinical diagnostic criteria for vascular dementia are still argued, even from a neuropathological perspective. Nowadays, we accept the generic definition of genetic vascular dementia (CADASIL or CARASIL), macrovascular dementia (multi-infarct dementia or strategic infarct dementia), or microvascular dementia (subcortical vascular dementia or more appropriately, small vessel disease-related dementia) [13,14,15]. Very recently, DSM V [16,17] and the STRIVE Consortium (Standards for reporting Vascular changes on Neuroimaging) conditioned the diagnostic criteria to specific neuroimaging studies [5,18]. In particular, the diagnostic criteria for the small vessel disease should include, in a conventional MRI, recent subcortical infarcts, white matter hyperintensities, lacunes, prominent perivascular spaces, and cerebral microbleeds [5,18]. Therefore, we take small vessel disease (SVD) into account, which is the consequence of the different damages to the small penetrating arteries and arterioles in the pial and lepto-meningeal circulation, along with penetrating and parenchymal arteries and arterioles, pericytes, capillaries, and venules [19]. SVD prevalence increases exponentially with aging. Around Europe, the prevalence rates of SVD related dementia, between ages 65–69 to 80+ years, ranged from 2.2 to 16.3% [20,21,22,23]. As aforesaid, aging is the most critical risk factor in developing the small-vessel disease, due to the loss of arterial elasticity, and a consequent reduction of arterial compliance [24]. The loss of arterial compliance is the major determinant of the altered autoregulation capacities, which leads to the deep sensitiveness of the brain of SVD patients to brisk decreases of blood pressure [25,26,27]. Moreover, apart from the reduction of elasticity, it should be considered that aging also causes a low-level functioning of the autonomic nervous system, with direct and endothelium-mediated altered baroreflex activity [28,29,30,31]. Pathological expressions of SVD are the arteriolosclerosis process and cerebral amyloid angiopathy (CAA) [32,33,34,35]. After that, even if debated, SVD could affect the integrity of the medial cholinergic pathway, for the hypoperfusion preferred localization, in the deep white matter capsule, [36], or, due to the multiple lacunar infarcts, the basal forebrain cholinergic bundle could be deafferentated from the tubero-mamillary tracts [37,38]. These aspects affect the normally-accurate cerebral flow regulation and they can further disturb the “retrograde vasodilatation system” with necessary consequences in neurovascular coupling [39]. Cerebral small vessel disease includes a neuroimaging and pathological descriptions, which comprise different imaging changes in the white matter and subcortical grey matter, including small subcortical infarct, lacunes, white matter hyperintensities (WMHs), prominent perivascular spaces (PVS), cerebral microbleeds (CMBs), and atrophy. Moreover, an associated hypoperfusion progression characterized SVD, causing incomplete ischemia of the deep white matter [7,40,41,42] accompanied by inflammation, diffuse rarefaction of myelin sheaths, axonal disruption, and astrocyte gliosis [35]. In small vessel disease, the occlusion of deep periventricular-draining veins is also evident [43], together with a disruption of the blood-brain barrier; the two factors together causing a severe leakage of fluid and plasma cells to potentiate the inflammatory cascade, which seem to happen in the course of chronic hypoperfusion, by collecting multifactorial causes for white matter alterations [44,45,46]. Cerebral small vessel disease is what has been described as “a progressive disease” [35]. Lesions progress over time, and the long-term outcome and impact on brain damage vary, even not knowing why or how; reasonably, it should be said that the most rapid and confluent progression of the isolated white matter hyperintensities could be considered as being the most relevant, to the best of knowledge, predictor of the fatal progression of SVD [47,48,49,50]. Of course, the total amount of lacunes and profound white matter alterations relate to the degree of cognitive impairment [51,52,53]. The preferred location of the lesions is placed along with the frontal and prefrontal-thalamus-basal forebrain networks, [54,55], directly implying the so-called cortical-deafferentation. Additionally, lesions due to SVD are specific to the caudate nucleus (the most precociously affected region), the putamen, insula, precentral gyrus, inferior frontal gyrus, and middle frontal gyrus. The higher metabolic request of these regions (more than 20%) at steady state than other brain areas fully explains the pathology [56,57,58,59,60,61,62,63]. On the other hand, SVD usually implies a reduced metabolic rate of oxygen (estimated of about 35% in white matter) [64,65]; metabolic incongruity between the brain oxygen supply and its consumption has been described in SVD, which determines an altered neurovascular coupling and altered vasomotor reactivity [35,66,67,68,69,70,71]. Neuropsychological pattern profiles of dementia that are related to SVD are related to the subcortical-cortical loops deafferentation and they are distinguished by poor executive function, poor planning, working memory alterations, loss of inhibition, reduced mental flexibility, multitasking procedures invalidation, and decrease speed of executive process [72,73,74,75,76,77,78]. Any specific treatment has been discovered, either as pathogenic or highly standard recommended for this condition.

Insert Figure 1 appr. here:

## 3. Anatomical and Structural Weaknesses in Small Vessel Disease

SVD is considered to be the major contributing factor or the sole responsible for the “generic defined” dementia-syndrome worldwide [79]. The small vessels represent its principal target, which include pial and small penetrating arteries, small intra-parenchymal arterioles (with smooth muscle cells), perivascular spaces, astrocytic endfeet, cerebral capillaries and veins, and venules. There is wide speculation on all the structures involved, to establish a potential role in the development of the chronic ischemic-hypoxic state, which is the final responsibility for the SVD, even if the principal SVD-model is the arterioles damage-based, and even if we do not know much about perivascular spaces. The pathophysiological role of PVS, their function and interaction with cerebral microcirculation, has not been established yet. There is a broad consensus that PVS forms a network of spaces around cerebral microvessels, acting as a canal for fluid transport, the exchange between cerebrospinal fluid (CSF), and interstitial fluid (ISF) and the clearance of catabolites from the brain. The perivascular compartment contains several cell types, like perivascular macrophages, pial cells, mast cells, nerve fibers, and collagen fibers [80]. Usually, as arterioles penetrate deeper into the brain, the glial membrane, the pericyte membrane fuse together and then obliterate the perivascular spaces [80,81], but it has been proposed that either in humans either in animals, the perivascular space could act as a brain lymphatic system, also being defined as “para-arteriolar”, “para-venular”, “paravascular”, or “glymphatic” [82]. This system has many complex functions (further in the review, we will explain it regarding neurovascular coupling), but it seems likely to exert the drainage work of the brain. Therefore, modification of this system produces deleterious effects, whose results are an accumulation of catabolites and toxic substances, together with a pronounced neural starvation [83,84]. In SVD, this system is invalid; one of the SVD hallmarks is the enlargement and widening of PVS, due to an obstructive process that is maintained by catabolites, proteins, and cell debris [82]. In small vessel disease, the occlusion of deep periventricular-draining veins is also evident [43], together with the disruption of the blood-brain barrier (BBB). All these facts together lead a consequent leakage of fluid and plasma cells, which eventually might potentiate the perivascular inflammation, and all of the cascades of the inflammatory/obstructive/stagnation-induced process [44,45,46,85]. The immobility of the fluid drainage can support PVS’s role in different diseases: the possible explanation of the PVS involvement in SVD, is the argued relationship demonstrated between an altered cerebrovascular reactivity (CVR), which is the change in cerebral blood flow in response to a vaso-active stimulus in the so-called neurovascular coupling, the found BBB dysfunction, and the correspondent perivascular inflammation [86]. Therefore, a lacuna should not indicate an enlarged perivascular space, as it is, still nowdays; it should never be the correspondent of the CSF-filled cavities on brain MRI or residual lesion of a small hemorrhage [82,87,88,89,90,91,92]. Nowadays, it should be more appropriate for the definition “lacuna of presumed vascular origin” to replace the term ”lacuna” [20,93,94,95,96]. 

### 3.1. Arteriolosclerosis as a Functional Model for SVD

Arterioles are the best studied target for SVD, starting from the pathological process that they undergo, the arteriolosclerosis. Arteriolosclerosis occurs in two primary histological forms, the hyperplastic and the hyaline arteriolosclerosis [97,98]. The hyperplastic is the most common lesion, principally due to the chronic state of hypertension. It begins with the hypertrophy of the smooth muscle in the media, and it is accompanied by the reduplication of elastic laminae, the growth of new cells in the intima, and the deposition of collagen, which progressively substitutes the muscle cells (onion skin arteries) and severely obliterates the lumen [97]. Hyaline sclerosis is another change in the vessels of hypertensive patients: the vessel wall becomes thickened with collagen [99]. Arterioles undergo a progressive deposition of hyaline material throughout the entire circumference of the vessel and which extends through the media [100]. The hyaline material is a consequence of the leakage of the plasma proteins, mainly the inactive form of complement (C3b) through the endothelium, and also by an increment of the basement membrane components by the smooth muscle cells [100]. Healthy aging implies the loss of the Windkessel effect and the loss of arterial elasticity, which reflects an anticipated and precocious return of the so-called wave reflection. Healthy aging also determines an increase of the systolic and a decrease of diastolic pressure, with a loss of resting flow effect through the Willis, which decrements the usual high perfusion pressure towards the most profound small arteries of the brain [101,102,103], thus provoking a loss of brain flow autoregulation. Arteriolosclerosis perpetuates the hypo-perfusion in the profound territories that are irrigated by penetrating arteries.

### 3.2. Hypoperfusion and Neuroinflammation

It is intriguing enough that chronic cerebral hypoperfusion defeats the traditional and acknowledged way of the anatomical thinking, with regards to the preponderance of cortical neurons on the other brain structures. Nowadays, it is well accepted that 10 min. of transient global ischemia in rat-brains determines a precocious sufferance of the perineural spaces, and then of the white matter, along with the internal and external capsule; one day after the ischemia, oligodendrocytes die [104,105,106], and the neuronal death occurs several days after the initial damage [107,108]. Moreover, as experimentally demonstrated, ischemia occurs in the brain (rat, mouse, and rabbit) [109,110], it seems evident that there is an induction of significant microglial activation, with significant regional variability [109]. When measured the time of onset, it has been described that microglial activation firstly appears in the hippocampus, but the activation does not last more than 48 h [109]. In the meantime, from 48 to 72 h after the ischemia, there is increased activation of the microglia. It occurs throughout the white matter, and the thalamus (from the second day after up to the fourth day). From the fourth day, the activation occurs through the cortex, protracting until 30 days after the initial ischemia [111]. Besides, microglia tend to retract their branches after ischemia, leading to a reduction in the total length and the total number of microglial processes [112]. The loss of blood flow in the peri-infarct region results in marked de-ramification and amoeboid transformation of soma [113]. Microglial activation is believed to be involved in the pathological progression of ischemic tissue. However, the function of activated microglia in ischemic events remains not entirely understood [113,114,115]. The experimental models seem to validate the hypothesis of two-step sequential microglia activation: the first one, mostly dependent on M1 type activation, with the production of oxidative species, proinflammatory cytokines, and lysosomal cascades [116]; soon after, there is an M2 activation, which seems to be reparative and blocking the inflammatory cascade of events [113,116]. It has been supposed that when the ischemic event is not an acute one, but there is chronic ischemia, like in SVD, there is a preponderance of M1 activation, with minimal M2 action [117,118,119,120]. Chronic ischemia determines a severe oligodendrocyte degeneration; soon after, it causes microglial activation and it is further associated with an increase of apoptosis processes that are associated with an elevation of caspase 3 RNA, and of matrix-metalloprotease 2 (MMP-2) expression [121,122]. Astrocytes react to the chronic ischemic condition, as a result of the length and severity of the insult. In the early ischemic period, the astrocytes respond with a remarkable proliferation, but, in the case of persistent hypoperfusion, with their degeneration and death [123,124,125]. It has been argued that astrocytes act in response to the ongoing modification of the neurons metabolic changed requests, possibly through glutamate signaling [126], and contribute to the regulation of blood flow modifying capillary permeability, by stretching out their endfeet to the microvessel, establishing a proximal connection with the capillary. Their death, due to chronic hypoperfusion, leads to an expanding, and auto-potentiating system of neuronal death, due to a misleading neurovascular coupling. The deterioration of astrocytic function at the late stages of white matter hyperintensities also supports the progressive character of SVD, as shown in a recent clinicopathological study [127]. The more actual histological works discovered collagenous pouches and tubes around small vessels, now referred to as vascular bagging, suggested as a possible biological marker of SVD [128]. Ultrastructural studies have found the splitting, branching, and thickening of the capillary basement membrane and perivascular deposition of collagen, also called microvascular fibrosis, in the brains of aged rats [129] and rhesus monkeys [130]. Frosberg et al. [128] showed vascular bagging in the frontoparietal and temporal control deep white matter. The Authors found that plasma proteins fill the vascular bagging, and argued that SVD should be characterized by a porous endothelium and an altered basement membrane [128,131]. Frosberg et al. [128] showed that, in SVD with diffuse white matter alteration, smaller basal ganglia vessels, including pre-capillary arterioles and capillaries, revealed vascular bags with COLL4-positive walls [132,133,134]. Post-capillary venules also showed vascular bagging in SVD, but they cannot be distinguished from capillaries, merely based on vessel diameters, deformed erythrocytes squeezing through vessels, or the presence of pericytes [135]. While the pericytes that were found in the Frosberg et al. work [128] were located outside the vascular bags, these cells and their processes are enclosed by two layers of the basement membrane [130], and therefore their degeneration might contribute to a splitting of the basement membrane, supporting what we have afore-described. Frosberg et al. [128] also reported the presence of string vessels in SVD. String vessels are thin connective tissue strands, remnants of capillaries, with no endothelial cells and without the primary function of blood transport [129]. String vessels suggest the precise location of the originally normal-functioning vessels, and after significant events (abrupt or chronic ischemia, aging, but also neurodegenerative disorders), they gradually disappear [136]. Many events induce their regression, which is probably due to a converging two-vias: an induced apoptotic phenomenon associated with the destruction of endothelial cells, attached by macrophages. Frequently the regression might be triggered by the loss of the vascular endothelium grogth factor (VEGF) [136]. In their work, Frosberg et al. [128] put in evidence four types of string vessels, suggesting different stages of string vessel formation, and an enhanced density of COLL4-positive string vessels and ghost vessels that resembled remnants of string vessels [128]. Quite suggestive is the finding that the higher quantities of string vessels that are described in work [128] have been found in the damaged white matter parenchyma. However, in some cases, the Authors have found that after an endothelial death, the empty basement membrane tubes could help the regrowth of new endothelial cells, which can synthesize new basement membrane layers [137], which could give the reason of the multi-layered vascular bags that are found in the original work [128].

### 3.3. Cholinergic Role in Small Vessel Disease

Moreover, small arteries undergo a systemic poorness of cholinergic network regulation. Many hypotheses have been raised for a possible explanation, starting from an altered cholinergic response to inflammation, which is a constant in chronic ischemic condition [138,139,140,141], up to a disruption of the cholinergic networks, which subcortically approaches the basal forebrain, since this is a preferential location of lacunar vascular infarcts and chronic hypoperfusion syndrome [142,143,144,145]. It has been widely demonstrated, either in animal models either in postmortem studies, that there is a reduced level of acetylcholine (Ach) in patients with vascular dementia [146,147], both in the cortical areas, in the hippocampus, and the cerebrospinal fluid [148,149,150,151]. A loss of cholinergic neurons in 40% of demented vascular patients was reported, accompanied by reduced ACh activity in the cortex, hippocampus, and striatum [152]. Post-mortem SVD studies revealed lower choline acetyltransferase (ChAT) activity when compared with the controls [153], and SVD patients have lower CSF concentrations of Ach [151,154,155,156]. In fact, in the experimental condition, the selective muscarinic antagonism by atropine, for example, has dramatic consequences in the CA1 region, [157]. Other experiments demonstrated that the selective alpha-7-nicotinic AChR antagonism exacerbates the hypoxic effects on the CA1 and CA3 cortical areas; on the contrary, non-selective nicotinic AChR antagonists have a detrimental effect on the hippocampus, not in all the other cortical areas [158,159,160,161]. The chronic reduction of the cerebral blood flow can affect the control of the cholinergic networks, but it happens that a proper cholinergic function is compulsory to well-regulation of the regional brain blood flow [162,163]. Ach principally mediates the parasympathetic innervation of the Willis circle and the pial vessels [164]. Ach stimulates in vitro the arterial relaxation, directly and via the promotion of the synthesis of vasodilator endothelium agents [164], via the nitric oxide synthase [165] and the GABA interneurons [166,167,168]. The stimulation of the Nucleus Basalis of Meynert results in increased blood flow throughout the cerebral cortex in experimental animals [169]. Upon stimulation, perivascular cortical afferents release Ach into endothelial M5 muscarinic receptors [148,170]. M5 receptors are highly expressed in blood vessel walls [170]. Yamada et al. [148] prepared knockout mice (M5−/−) and found that, as compared to wild-type mice, these animals lose the ability to dilate cerebral arteries, but could still regulate extra-cerebral flow. Upon stimulation, perivascular cortical afferents release Ach onto endothelial M5 muscarinic receptors [170,171]. Hamner et al. [172] demonstrated that cholinergic control of the cerebral vasculature might be active at low frequencies, lower than 0.05 Hz when the sympathetic nervous system appears to play a role in the cerebral auto-regulation, in a limited but well-conducted study. At these low frequencies, the myogenic mechanisms appear to play any role; to surprise, the correspondence between cholinergic and sympathetic cerebrovascular regulation above 0.05 Hz is striking, suggesting that the cerebral circulation engages different mechanisms to protect itself [172]. There are many different reasons for the cholinergic impairment that was observed in SVD. The cholinergic impairment for artery dysregulation that was observed in small vessel disease could derive from the deafferentation of the basal forebrain cholinergic bundle to the subcortical structures, due to the most probable location of lacunar vascular events and the chronic hypoperfusion consequences, aforementioned [36,38,173]. Bohnen et al. [36] demonstrated in vivo that the severity of the periventricular white matter lesions is associated with lower AchE activity, in the middle-aged and elderly subjects without dementia, as a result of cortical cholinergic deafferentation. In animal SVD models, there is a concomitant reduction of vasopressin and histamine, which is interpreted as a result of the interruption of the tracts that comes from the supra-optic and tuberomammillary nuclei and ends in the basal forebrain [174]. The reduction of vasopressin and histamine seems to have a redundant effect on hypoperfusion. Some clinical data confirm a reduction of the number of cholinergic neurons in the Nucleus Basalis of Meynert in multi-infarct dementia, but not in SVD [175,176,177]. It has been conveyed that a primary loss of the cholinergic neurons of Nucleus Basalis of Meynert does not mediate cholinergic impairment [178,179], but is a consequence of the secondary cholinergic deficits, due to the indirect, cholinergic endothelial effect, aforementioned. Though, the number of muscarinic cholinergic receptors is markedly reduced in mixed dementia patients [179] and SVD dementia. Cholinergic poorness promotes a less efficacious endothelium relaxation, even due to an altered nitric oxide synthase and loss of efficacy of the GABA interneurons [165,166]. The two mediators seem to be less efficient in influencing the small arteries contraction [180,181,182,183]. A final step on this point has been written by a probabilistic tractography analysis [155]; this study tracked the two primary white matter tracks which map to cholinergic pathways, identifying a significantly lower fractional anisotropy in precocious form of SVD. Mediation analysis demonstrated that fractional anisotropy in the tracked pathways could fully account for the executive dysfunction, and partly mediate the memory and global cognition impairment. The recently published study [155] study suggests that the fibers that are mapped into the cholinergic pathways, but not those of the Nucleus Basalis of Meynert, are significantly damaged.

Finally, an alteration of the conceptualized “cholinergic anti-inflammation pathway” summarized another possible cause for the cholinergic poorness [138,139]; these findings are based on the knowledge that acetylcholine released from cholinergic axon terminals can interact with α7 nicotinic Ach receptors on vicinal immune cells. The nicotinic receptors then translate the cholinergic signal into the suppression of cytokine release, being involved in the inflammatory cascade [140,141]. A chronic proinflammatory condition counterbalances the acetylcholine release and promotes its cascade effects on the vasoregulation. The pathological cascade of events, which occurs as a consequence of all the pathological alterations described, determines a decrease of the vascular tone, with a release of the blood-brain barrier permeability, with a loss of the internal vascular remodeling and with major vascular rarefactions. As a result, hypo-perfusion at rest occurs in the brain and it is associated with impairment in the moment-to-moment control of CBF, with a decrease of adaptive vascular responses and with a diminishment of the neurovascular coupling and auto-regulation system [145,180].

Insert Figure 2 approx here:

## 4. Chronic Hypoxia and Brain Response

The brain requires a disproportionate amount of the body’s energy. Up to 20% of cardiac output is devoted to meeting the brain’s energy demands, despite accounting for only 2% of body mass [184]. The cerebral vasculature possesses well-developed mechanisms that enable cerebral blood flow (CBF) to remain constant during fluctuations in arterial pressure (autoregulation) and meet the increased nutrient demands when local brain activity rises in order to deliver these nutrients effectively and protect the brain from hypoperfusion and ischemic damage [185,186]. Cerebral SVD significantly and chronically impairs the ability of the cerebral vasculature to meet these demands due to several structural and functional changes, which ultimately result in brain injury, cognitive decline, and dementia. Cerebral vasoconstrictor and vasodilator responses are important mechanisms by which brain blood flow is maintained, as aforementioned. In SVD, chronic hypoperfusion leads to a decrease in cerebral blood flow, hypoxia, oxidative stress, and triggers inflammatory responses, which leads to a potentiated hypoperfusion condition. The induced lesions are mostly expressed in the white matter (WM) and especially in the periventricular WM, basal ganglia, and hippocampus. Hypoxia-induced oxidative stress leads then to mitochondrial dysfunction, neuronal damage, and apoptosis via the nitric oxide synthase (NOS) pathway [187,188,189,190]. Chronic hypoxia profoundly influences vascular control, altering both vasoconstrictors as well as vasodilator responses in isolated cerebral vessels; indeed, chronic hypoxia alters the contractile response of the isolated cerebral vessels [191]. Chronic hypoxia is known to influence Nitric Oxide (NO) modulation of contractile response. In one animal study, the authors showed that chronic hypoxia augmented contractile sensitivity to the thromboxane mimetic U-46619 in isolated cerebral vessels as the result of reduced nitric oxide (NO) production and activity. A decrease in NO production of L-arginine and oxygen increased NO degradation or reduced cyclic guanosine-3-5-monophosphate (cGMP) production (involved in smooth muscle relaxation). In this case, the administration of the nonspecific NO synthase (NOS) inhibitor nitro-L-arginine (NLA) eliminated the difference in contractile sensitivity between the vessels from the normoxic and chronically hypoxic animals, which suggests that a reduction in NO production and activity was responsible for the increased contractile sensitivity that was observed [192]. Such effects may be significant in the adult in whom disorders involving cerebral circulation occur under conditions of acute and chronic hypoxia. Chronic cerebral hypoperfusion (CCH) is a prevalent pathophysiological state in patients with Alzheimer’s disease (AD) and vascular dementia (VaD). CCH has been identified as one of the initial conditions that are critical in the development of cognitive dysfunction [193]. In several studies, deranged energy metabolism, glial activation, apoptosis, oxidative stress, neuronal damage, and white matter lesions that are caused by cerebral hypoperfusion have been found to contribute to the pathophysiological mechanisms that lead to cognitive impairment [194,195]. Animal models of CCH showed that such compensatory actions induce abnormal activation of the frontal cortex and the hippocampus. Hypoxemia, in addition to hypoperfusion, exacerbates ischemic brain damage and it is associated with more severe white matter lesions. Abnormal cerebral hypoxia induces compensatory and adaptive mechanisms to prevent hypoperfusion injury and preserve recovery of brain function [194,196]. Part of those adaptive mechanisms involves increased capillary diameter, neovascularization, and enhanced expression of vascular endothelial growth factor (VEGF). In the condition of CCH, hypoxia-inducible factor 1 (HIF-1) is one of the most important transcription factors that are involved in the endogenous adaptive response. HIF-1α then leads to the expression of a large number of genes. It regulates more than 2% of the genes in human vascular endothelial cells [197] and is recognized today as a regulator of the vast majority of hypoxia-inducible genes that are responsible for the cell adaptation to hypoxia, including angiogenesis, anaerobic metabolism, mitochondrial biogenesis, erythropoiesis, vasomotor control, and cell proliferation, such as vascular endothelial growth factor (VEGF), glucose transporter-1 (GLUT-1), and erythropoietin (EPO), all factors that lead to survival under hypoxic conditions [198,199]. HIF-1a is also involved in hypoxia-dependent inflammation, apoptosis, and cellular stress. Animal models showed that the neuron-specific knockdown of HIF-1a aggravates brain damage after a 30 min. middle cerebral artery occlusion (MCAO) and reduces the survival rate of those mice, and an impairment of learning and memory after four weeks of CCH has been reported. Cerebral angiogenesis is reduced, while oxidative damage is also promoted with the proliferation of astrocytes and microglia in the cortex and some sub-regions of the hippocampus [200]. In other studies it is reported that the lowering of oxygen induces hypoxia-inducible factor-1α (that is involved in neuroinflammatory response), which has the direct consequence of the hyper-production of free radicals and proteases, BBB disruption, vasogenic edema, and myelin damage; all these effects may lead to white matter (WM) damage and vascular cognitive impairment. Moreover, hypoxia-induced MMP-9 expression leads to vascular leakage, which MMP inhibition could reduce. Pharmacological blockage of MMP-9 or MMP-9 gene deletion confers neuroprotection in traumatic brain injury and stroke [201].

Protective mechanisms that are triggered by hypoxia are characterized by decreasing the O2 demand, increasing the O_2_ supply, or a combination of both. Some animals can reduce the O2 demand through a condition called hypometabolism, but, in the human brain, this condition is poorly expressed. Hypoxia is always associated to early signs of failure that are represented by marked falls in pH and tissue creatine phosphate levels, followed by a dysfunction of Na+/K+ ATPase and lethal ion imbalance [202,203]. In the human brain, pro-survival pathways and improving brain oxygenation actions are activated. During cerebral hypoxia, in brain, HIF-2, also known as EPAS-1 (endothelial PAS domain protein 1), is expressed, principally in endothelial cells, including brain capillary endothelial cells [204]. HIF-2 is active during prolonged mild hypoxia and it might be involved in brain microvascular response. In one paper, the authors provided evidence that HIF-mediated pro-survival responses are dominant in rats with CCH. The activation of HIF-1 is part of a homeostatic response that is aimed at coping with the deleterious effects of CCH [200]. While considering these premises, a large number of clinical trials tried to identify protective strategies against cerebral impairment after hypoxia through the identification of endogenous neuroprotective pathways. Based on animal work, it has been shown that spontaneously hypertensive/stroke-prone rats (SHR/SP) with unilateral carotid artery occlusion had white-matter damage while being treated with a permissive Japanese diet. One week after, white matter showed a significant increase in hypoxia-inducible factor-1α (HIF-1α), which increased further by three weeks. The BBB disruption was supposed to be secondary to hypoxia and related to a matrix metalloproteinase-9 (MMP-9)-mediated infiltration of leukocytes. In those animals, treatment with minocycline significantly reduced the lesion size and improved cerebral blood flow. Minocycline prolonged survival [205]. The results are far from to be applied in the human chronic hypoperfusion condition for the aforementioned cascades of events that appear to be determinant in human SVD.

Insert Figure 3 approx. here:

## 5. Endothelium and SVD

The brain endothelium, even in severe SVD (presenting an almost complete loss of myocytes and other mural cells) remains intact, even if the endothelium is one of the main targets of the redox altered process and inflammation (and both these processes are highly activated in SVD) [134,206,207,208,209]. This paradoxical survival of the brain endothelium is also evident in patients with CADASIL [206,207]. On the contrary, systemic endothelium activation is quite different in SVD. 

Thus, indirectly, brain endothelium suffers in SVD conditions. Mitochondrial senescence of the endothelium walls has a catastrophic effect on cerebral endothelial cells [210]; this alteration, which is over-expressed in SVD [211], is generally related to an impaired response to the three major endothelium-derived nitric oxide-vasodilators [212], prostacyclin [213], and endothelium-derived hyperpolarizing factors (EDHF) [214]. The reduction of NO production is derived from an impairment of the mitochondrial functions, being caused by a hyperproduction of the anti-oxidative defense system, and an increased O2 anions reaction with NO, producing peroxynitrite [215]. The activity of endothelial NO synthase (eNOS), which catalyzes the production of NO, declines with aging [216], but is even more impaired in SVD, where an important downstream target of Rho is the Rho-associated protein kinase (ROCK) [217]. These ubiquitously expressed serine/threonine protein kinases are involved in diverse cellular activities, including apoptosis, smooth muscle contraction, cell adhesion, and remodeling of the extracellular matrix [218]. In the regulation of endothelial cell, migration ROCK interacts with ezrin, radixin, and moesin (also known as the ERM proteins) that function as cross-linkers between the plasma membrane and actin filaments [217] and are indispensable for the leukocyte adhesion molecules coordination, being essential for barrier function [219]. Moreover, the ROCK/RhoA complex regulates the eNOS, as previously exposed [217]. NO-induced vasodilation occurs via the activation of myosin light chain phosphatase (MLCP) in a cGMP dependent manner. RhoA/ROCK counteracts this through MLCP inactivation and calcium desensitization [217,220]. ROCK/Rho decreases eNOS expression and affects the availability of NO [221]; it has also been proven in brain small vessels, even if these effects have been largely studied in major vessel disease (coronary) [82]. Three potentially functional eNOS polymorphisms (T-786C, intron 4ab, G894T) located toward the 5′ flanking end of the gene are known to be considered as being present in SVD and also in isolated lacunar infarction and ischemic leukoaraiosis [222]. RhoA inhibition overwhelms VEGF-enhanced endothelial cell migration in response to vascular injury, without, or better said, with a minimal effect, on basal endothelial cell migration [223,224]. The maintenance of the endothelial barrier is a prior role of the endothelium cells, mainly through the operative system of RhoA [225], also being mediated through the regulation of Vascular endothelium cadehrins (VE-cadherins) [226]. In diabetes (one of the main risk factors associated to SVD), advanced glycation end products (AGEs) accumulate in the vasculature, triggering a series of purposeful and morphologic changes of endothelial cells, such as the increase of the activation of the RhoA/ROCK pathway; the significant consequence is an increased endothelial cell permeability [227]. It can also act as a VEGF inducer, which indirectly causes microvascular endothelial hyper-permeability [228]. 

Therefore, it should be argued that the endothelium seems to be functionally impaired in SVD, even if morphologically and structurally undamaged [229]. 

The endothelial NO downregulation in SVD is a marker of decreased endothelial regulatory capacity, in response to external stimuli, such as hypercapnia [230,231]. Living studies have demonstrated a significant baseline CBF reduction in SVD–affected subjects, together with an impaired CBF autoregulation [232,233,234]. Endothelial activation refers to the change in the expression of many different surface markers [235,236,237,238]. These circulating markers of endothelial activation include intercellular adhesion molecule-1 (ICAM-1), which has been considered as a generic expression of white matter progression [239], soluble thrombomodulin (sTM), interleukin-6 (IL-6), plasminogen activator inhibitor-1 (PAI-1), von Willebrand factor, and others [207,240,241,242]. Moreover, an upregulation of hypoxia-endothelial-related markers has been proven, such as HIF 1 alpha, VEGFR2, and neuroglobin, when white matter lesions appear to be confluent [243]. The matter is even more impressive when it appears evident that endothelium in overall activated, as described above, but, according to some authors, not specifically in the human gray matter [209,241,244,245]. Though, the brain endothelium NO dysregulation implies not only a direct inhibition of the vessel tone, but indirectly, more critically, a decrease of the dynamic neurovascular control mechanism [246,247]. 

Moreover, the permanent status of oxidative stress-induced should be taken into account, which causes a superimposed macroscopic alteration of the cerebral endothelium. 

The immediate consequence of the endothelial dysfunction has two significant consequences, the reduction of the resting flow in the marginally perfused white matter and macroscopic alterations of the BBB permeability [247]; these two aspects lead to additional oxidative stress, by inducing tissue hypoxia and extravasation of the plasma proteins [247], and both of them potentiate the inflammation pathway, through the Nuclear Factor Kappa-Light-Chain-Enhancer of activated B cells (NFkBeta) dependent transcription. The modern view gives the endothelium the control role of the propagation of vasomotor signals [248], even if the question is still unresolved. In systemic vessels, the endothelium is well known to participate in the retrograde propagation of vascular signals [81,249], but in the brain the mechanisms by which endothelium interacts with the spread of the vascular signal is still debated. It has been proven that a highly localized lesion of the endothelium failed to propagate beyond the lesion site, and altered the amplitude and temporal dynamics of the go-ahead vascular sign, with weaker temporal coordination [250]. It has been demonstrated that brain endothelium is enriched with K_IR_ channels, and not by K_Ca_ channels; these channels are sensitive to high K flow, being derived from neural activity, and are transmitted by the synapses or by astrocytic end-feet [249,251]. It has been recognized that K^+^ is recognized in the endothelium, and the upstream penetrating arteriole is the effector of the vasodilatation [251], and its rapid propagation is probably conducted by ionic currents traveling through the endothelium via gap junctions and then through the myoendothelial junctions [249]. Therefore, K_IR_ suppression avoids the increase of CBF that is produced by cortical activation [251]. The most intriguing aspect of the endothelial conductance is the fact that the conducted vasomotor responses, either being a dilatation, either vasoconstriction, can be generated by different neuromodulators, i.e., Acetylcholine, ATP, prostaglandin F-2alpha, and NO, but their effects on neurovascular coupling has never been determined [252]. The evidence is increasing on pial arterioles: signals that are generated by the neuronal activity, deep in the brain, should be conveyed to upstream arterioles, remote from the area of activation, to increase flow efficiently [249]. Vascular mapping and fMRI demonstrated that vascular responses are first seen in the deep cortical lamina during somatosensory activation, and then, more superficially, suggesting a retrograde propagation of the vascular response [253]. A possible scenario for the transmission and coordination of the vascular response is described [249], as follows: activation-induced increase in extracellular potassium triggers the hyperpolarization of capillary endothelial cells and pericytes [254]. The hyperpolarization propagates upstream and reaches smooth muscle cells in penetrating arterioles, producing relaxation [249,251]. At the same time, metabolic modifications (reduced viscosity, increased deformability of blood cells) on the endothelium of feeding arterioles increment the smooth muscle cell relaxation (the so-called flow-mediated vasodilation). In upstream pial arterioles, remote from the site of activation, there is vasodilation, by propagation from arteriole downstream and acting as a local flow-mediated and myogenic response. For all of the conditions mentioned above, SVD is defective in neurovascular coupling, even for endothelial and pericytes failure.

## 6. Astrocytes and SVD

Neurons, astrocytes, oligodendrocytes, as well as vascular and perivascular cells, are intimately related to metabolic control and they act as trophic determinants in brain development, function, and reaction to injury. Specifically, astrocytes play integral roles in the formation, maintenance, and elimination of synapses in development and disease [255]. In addition to their well-established interactions with neurons, astrocytes are also needed for the development and maintenance of BBB characteristics in endothelial cells [256], and for the reorganization of vascular networks after brain injury [257]. In turn, the endothelial cells regulate glycolytic metabolism in astrocytes through the production of NO [258]. The release of vasoactive substances, such as prostanoids from astrocytes, can couple cerebral blood flow to neuronal energy demand, and astrocytes supply neurons with vital metabolites, such as lactate in response to neuronal activity [259]. Additional homeostatic functions of astrocytes include water, ion, and glutamate buffering, as well as tissue repair after insult or injury [260,261]. The reactive astrocytes can release a wide variety of extracellular molecules, including inflammatory modulators, chemokines and cytokines, and various neurotrophic factors. These factors can be either neuroprotective (e.g., cytokines, such as interleukin-6 [IL-6], and transforming growth factor-b [TGF-b]) or neurotoxic (such as IL-1b and tumor necrosis factor-a [TNF-a]) [262]. The process of glial scar formation exemplifies the interplay between the neuroprotective and neurotoxic effects of reactive gliosis. The glial scar serves to isolate the damaged area and it prevents the damage extension by restricting the infiltration of inflammatory cells. However, molecules that are secreted by reactive scar-forming astrocytes can also be refractory to neurite growth [262,263]. An example of what the above written can be found in AD, where reactive astrocytes are found in the postmortem brain of affected patients [264,265,266]. It has been demonstrated that astrocytes can internalize Amyloid-beta plaques, exerting a scavenger-like function [267,268]. Nonetheless, it has been proved that those astrocytes with amyloid-beta are irreversibly compromised, likely showing altered calcium homeostasis [269]. Moreover, it is believed that astrocytes in AD are seriously compromised through the altered expressions of connexin 43, a hemichannel fundamental in astrocytic gap junctions, which is fundamental for astrocytes connections to neurons [270,271,272,273,274], through widespread modifications of energy metabolism and due to altered response to oxidative stress [274,275,276]. In SVD, there are some data [277] that concern the cerebral amyloid angiopathy. There is a decrease in the number of astrocytic processes contacting the vasculature in the cerebral cortex and hippocampus, which are even degenerated and undergo to the clasmatodendrosis process [278]. Price et al. [277] examined the end-foot changes of astrocytes in cerebral amyloid angiopathy and found out that there is a significant reduction of aquaporin-4-positive staining associated with the blood vessels, without any modification of the total protein or gene-expression [277]. Moreover, end-feet inward rectifying potassium channel, K_ir_ 4.1 and BK calcium-dependent potassium channel, being tightly co-anchored to aquaporin-4-water channels, have been found significantly decreased in mice with high levels of cerebral amyloid angiopathy [279,280]. Even though that these alterations are not indicative of a global decrease in potassium, as a whole, they can be considered as being a fundamental cause of altered astrocytic excitability, therefore causing an alteration in the neurovascular coupling [277,280]. Specific models of astrocytes impairment in hyperhomocysteinemic (HHcy)mice, which mimic small vessel disease, have been developed, with a three-step investigation (at 6, 10, and 14 weeks of B6, B9, and B12 detrimental diet in wild type HHcy mouse) [281,282]. These studies found out that after ten weeks on a diet (at the most after 14 weeks), end-feet disruption occurs. This phenomenon is concomitant to a reduced vascular labeling for aquaporin—4—water channels, lower protein/mRNA levels for K_ir_4.1 and BK potassium channels, associated with an higher expression of MMP-9 [277,283,284]. The most interesting finding is that in this mice model, microglial activation was evident since the precocious time of observation (six-week time), and precedes astrocytic changes [285,286,287,288].

Even if the data are not widespread, it could be said without approximation that astrocytes’ role, even if not at all understood, is seriously compromised in SVD, therefore interfering with autoregulation, repair process, and endothelium and BBB regulation.

## 7. Oxidative Stress in Angiogenesis and Vascular Disease

Hypoxia-induced oxidative stress leads then to mitochondrial dysfunction, neuronal damage, and apoptosis via the NOS pathway; it promotes a release of reactive oxygen species and free radicals [289,290]. Oxidative stress imbalances the ratio of antioxidants and reactive oxygen species, resulting in evident damage to vessel endothelial, glial, and neuronal cells, favoring a neurovascular uncoupling, and a global cerebral blood flow reduction [195]. The excess of reactive oxygen species further disrupts mitochondrial function and, thus, induces hypoxia and oxidative stress. Reactive oxygen species (ROS) are a group of oxygen-derived molecules that are generated by all mammalian cells. The conversion of molecular oxygen to superoxide by oxidase enzymes is the first step in the generation of all ROS. During physiological conditions, the production and metabolism of ROS are controlled by antioxidants to prevent cellular damage, and ROS may serve as important signaling molecules for the regulation of vascular tone. Thus, numerous conditions can disturb the oxidant/antioxidant balance inducing an increase of oxidants and oxidative stress. 

The reactive oxygen species that are produced during mitochondrial oxidative phosphorylation are associated with damage to DNA, lipids, and proteins. The accumulation of mitochondrial DNA mutations induces impairment of mitochondrial function. Mutations in mitochondrial DNA and oxidative stress both contribute to neuronal aging and neurodegeneration [291,292]. All of those changes contribute to neuronal loss. The effect of aging does not occur, to the same extent, in all brain regions. For example, when compared to cerebral areas, the cerebellum is protected from aging effects. Usually, the prefrontal cortex is most affected and the occipital least. Frontal and temporal lobes are most affected in men as compared with the hippocampus and parietal lobes in women [293].

ROS can be generated by all vascular cell types, including endothelial cells, smooth muscle cells, adventitial fibroblasts, and perivascular adipocytes. [294] There are two main endogenous sources in the vasculature: mitochondrial electron transport chain reactions and nicotinamide adenine dinucleotide phosphate (NADPH) oxidase [295,296] NADPH oxidase, an enzyme that generates superoxide anion by transferring electrons from NADPH to oxygen, is recognized as a major source of ROS in many cell types.

Cerebral arteries express several enzymes that are potential sources of ROS, including cyclooxygenase (COX), mitochondria, and the NADPH oxidases. Animal studies showed that NADPH oxidase activity and ROS generation are significantly higher in cerebral arteries when compared with systemic arteries in blood vessels from healthy animals (mouse, rat, pig, and rabbit) [297,298], suggesting an important role for ROS-dependent signaling in cerebral arteries, particularly important vasoactive molecules in cerebral arteries from healthy animals. The overproduction of oxidants might be responsible for oxidative stress; several enzymes, such as the NADPH oxidases, are responsible for this overproduction, which can be due to a diminished antioxidant enzyme activity or antioxidant levels. Moreover, NADPH-oxidase–derived ROS partly mediates flow-dependent responses of rat cerebral arteries in vivo [299]. ROS-mediated ischemic injury derives from the activity of NADPH oxidases, expressed in various cell types in the brain. The NADPH oxidase is a heteromultimeric enzyme complex with different separate subunits. Nox2-NADPH was the first NADPH oxidase to be revealed in neutrophils. Nox2-NADPH oxidase is a key mediator of ischemic brain injury, leading to BBB disruption, and the dysfunction of larger pial arteries, previous studies showed that Nox2-deficient mice have less brain injury after focal ischemia [300].

It is well proven that oxidative stress can damage cerebral vascular function via the disruption of endothelium-dependent NO signaling [301]. In animal models, different prototypes of sporadic SVD have been studied. Particularly, chronically hypertensive animals, chronic hypoperfusion models (carotid occlusion or stenosis models), and models of focal or global ischemia induced by surgical or pharmacological occlusion of a cerebral artery have been described. Those models implicate oxidative stress as a key mediator of ischemic brain injury [302]. Reducing NO bioavailability leads to lessened vasodilator, anti-proliferative, and anti-inflammatory properties. Additionally, this reaction generates other oxidant molecules, accentuating vascular dysfunction, damage to proteins and DNA, and reduced vascular responses to NO [303]. The loss of modulation of Rho-kinase activity is another critically important consequence of reduced NO bioavailability during disease. NO has been shown to modulate Rho kinase activity in cerebral microvessels, such that the inhibition of NO synthase activity increases the influence of Rho-kinase on vascular tone [304]. ROS may be able to directly increase Rho kinase signaling [181,305]. Besides, Rho-kinase can influence both eNOS expression (via effects on eNOS mRNA stability) and activity [306,307]. Therefore, a significant loss of NO promotes increased Rho kinase activity, which causes a decrease of NO synthase-derived NO, which in turn promotes Rho kinase activity. Other mechanisms of ROS-mediated vascular damage have been described during non-amyloid and amyloid cerebral SVD disease states, including the activation of poly (ADP)-ribose polymerase (PARP) and transient receptor potential melastatin-2 (TRPM2) channel activation [181].

Hypoxia and ROS induce pathological angiogenesis that promotes atherogenic processes by increasing both macrophage infiltration and the thickening of the blood vessel wall [295]. ROS and ROS-products directly promote angiogenesis, probably by a direct influence of the metabolites of lipid oxidation, which are generated and accumulated in plasma and the vessel wall [308,309].

The cycle of oxidative stress, endothelial and microvascular dysfunction, and inflammation exacerbate cerebral damage. The interplay between oxidative stress and inflammation primarily dysregulate the neurovascular unit, resulting in increased BBB disruption, edema, neurovascular uncoupling, and neuronal damage.

Recent data reported that genetic factors might be implied in SVD pathogenesis; similarly to what is described in Alzheimer′s Disease, the ApoE4 allele, genetic risk factor for sporadic AD, has also emerged as a risk factor for cognitive impairment, being mediated by vascular alterations [310]. ApoE4 increases the risk of WM lesions, independently of other risk factors, like age or hypertension. ApoE4 probably increases the production of reactive oxygen species. The mechanisms by which the ApoE4 may promote WM damage needs to be discovered. Healthy individuals carrying the ApoE4 allele have reduced CBF, and animals with the targeted replacement of ApoE with human ApoE4 have altered permeability of the BBB [311].

Potential treatment strategies for SVD might include those that target anti-oxidant effects for the endothelium of small cerebral vessels [181], as well as the endothelium of the BBB [312,313]. NO donors could be useful in releasing the functioning endothelium of small vessel disease, given that NO bioavailability is impaired in SVD. However, the clinical efficacy of traditional NO donors (e.g., glyceryl trinitrate) is limited by their susceptibility to tolerance development [181] and for a proved decreased effectiveness under oxidative stress [314]. Novel NO agents, such as nitroxyl (HNO) donors, show vasodilator properties, even in the presence of scavenging by superoxide and tolerance development, and it can suppress the activity of NADPH oxidase 2 (Nox2) in cerebral vessels [315]. Other potential interventions for improving artery function could include prostacyclin mimics, peroxisome proliferator-activated receptor gamma (PPARγ) activators, anti-inflammatory agents, phosphodiesterase-5 inhibitors, and Rho-kinase inhibitors.

The obvious strategy, like the administration of potent antioxidants, such as Vitamins C and E, has shown to be beneficial for vascular function in several experimental and small clinical trials [316,317,318]). However, disappointingly, the results of large clinical trials of antioxidant supplementation in vascular disease and stroke have largely failed to show any benefit. For example, the Heart Protection Study found no effect of Vitamin C, Vitamin E, and β-carotene supplementation on the incidence of vascular events (including stroke) in 20,500 high-risk individuals [319]. These results reflect the poor results that were obtained from the large trials for vitamin B12 and folic fortification [320]. The reasons could be the correct time of administration, the status of the beginning, and, above all, the knowledge that supra-physiological concentrations of Vitamins C and E would be required for competing with the reaction of superoxide and NO [316]. Enhancing the metabolism of superoxide by superoxide dismutase (SOD) could be an effective way of ameliorating the impact of this ROS on vascular function. All these conditions could be achieved by using native SOD that is modified to include a polyethylene glycol group, which improves its ability to enter cells and stability [181]. The ROS scavenger tempol is cell-permeable and it has been used in experimental studies [321,322]. Further studies are, of course, necessary for fully establishing their real therapeutic efficacy and the end of the therapeutic potential of ebselen (Gpx mimetic, Peroxynitrite (ONOO–) scavenger, and Nox inhibitor), and edavarone (O2-scavenger). Some clinical benefit has been reported for both ebselen and edaravone [323].

The central role of NADPH oxidase-derived ROS in cerebral SVD is strongly supported, being, therefore, that its inhibitors could be an effective strategy. It is well established that the Nox2 isoform is a major contributor to the entire NADPH oxidase activity. Therefore, this selective target is the most effective approach. Nevertheless, this isoform is also expressed in phagocytic immune cells; thus, it can be argued that prolonged selective therapies could help prevent SVD, but they invariably lead to an immunosuppression condition [324,325]. Notably, three of the most effective and frequently prescribed classes of drugs for the treatment of vascular risk factors have been shown to inhibit NADPH oxidases, reducing oxidative stress, are the Angiotensin-converting-enzyme inhibitors (ACE inhibitors), Angiotensin II receptor type 1 (AT 1) antagonists, and the statins [181,326,327]. Their primary roles, their potential side effects, and the eventual concomitant risks (hemorrhagic events in SVD caused by statins [328]) limit their employment as direct NADPH oxidase agents.

Diphenyleniodonium and apocynin are the most frequently employed NADPH oxidase inhibitors. Nevertheless, they have non-specific side effects that limit their clinical usefulness. For example, the flavin antagonist, Delta opioid agonist (DPI), inhibits all flavin-containing enzymes, including NO synthases [329] and cytochrome P450 enzymes [330]. Apocynin, on the other hand, can act as a pro-oxidant under certain conditions and inhibits the formation of prostanoids [181,331]. Some novel products that inhibit the Nox2-NADPH oxidase isoform are the gp91ds-tat [332]. Gp91ds-tat has proven to be an invaluable pharmacological tool in studies investigating the role of Nox2-NADPH oxidase in disease, but it has never been employed in population studies, due to its peptide formulation, which impedes the oral administration. Some other molecules, such as Triazolo pyrimidines (e.g., VAS2870 and VAS3947), are not isoform-specific and cannot be employed [333]. Pyrazolopyridine derivative (e.g., GKT136901 and GKT137831) inhibit Nox1-, Nox4-, and Nox5-NADPH oxidases, and they are orally bio-available, have a safe profile, and appear to lack significant off-target effects [334]. The flavonoid derivative S17834 was proposed to be an NADPH oxidase inhibitor that is based on its ability to inhibit superoxide production [335]. Ebselen and several of its selenium-containing analogs have been identified as inhibitors of Nox2 and Nox1 oxidase, with one analog (JM-77b) exhibiting potential Nox2 selectivity [336]. 

The small number of clinical trials, the selective time of enrollment, and more precise strategies of biochemical definition oblige referring to these therapeutic choices as future options, which will need stronger and affordable more extensive studies [337].

## 8. SVD: Inflammation as a Promoter or a Marker

Well defined, as afore-mentioned, is the inflammation role inside the brain as one of the principal factors of SVD. All of the principal mechanisms of inflammation, apoptosis, necroptosis, Wallerian degeneration, demyelination, astrocytosis, and microglial activation have been documented in SVD [338,339,340]. Necroptosis is a programmed cell death, not involving caspase but the loss of plasma membrane integrity, through the receptor-interacting serine/threonine-protein kinase (RIPK-1) and mixed–lineage kinase domain-like (MLKL); it has been documented in SVD, via an elevated presence of RIPK-1 [341,342], but also recently in vascular disease [343]. Wallerian degeneration is a cascade of events that includes the granular degeneration of the axonal cytoskeleton, accumulation of activated macrophages and microglia, and local changes, mostly in SVD [35]. Recently, even a potential outstanding site of priming effect has recently been determined, which is able to promote at a distance the neurodegenerative process, in the so-called microbiome-gut-brain axis. Priming infectious event might occur in the gut axis, and the subsequent release of microbiome bacteria, viruses, fungi, and protozoa, their toxins and their metabolites might promote the priming innate immune cells in the CNS [344,345,346,347,348,349]. Somebody wants to find a positive trivalent inter-relationship between neurodegeneration-inflammation-aging, by defining this as the “inflammaging” [350]. Aging has been associated with a low grade sterile inflammatory status of the immune system, in which Il6, TNF, and IL-1Beta are key players, being more evident in an unhealthy state. Neuro-inflaming can be considered as one of the most important etiological factors in age-related neurodegeneration [351], being associated to a reduction of neurons number, a decrease of neuronal arborization, and loss of spines [352].

What fits particularly in this theory is the strong evidence that, in an aging brain, both macrophages and microglia react with a prolonged and over activated response to stimuli [353]. This over-activation induces reactive oxygen species production, attracts peripheral leucocytes, and both of these conditions can activate glial cells [354]. The activation of the glial cells promotes telomere shortening, which can be a contributor for different neurological conditions, such as AD [355]. The contemporary impaired phagocytosis results in altered removal of toxic accumulations, such as those of AB 42 and alpha-synuclein. In SVD, the emerging role of PVS, astrocytic endfeet, cerebral capillaries and veins, and venules stands on their possible role of drainage system around cerebral microvessels, acting as a canal for fluid transport, the exchange between CSF and ISF and clearance of unwanted products from the brain [356]. Therefore, the modification of this system produces deleterious effects, whose results are an accumulation of catabolites and toxic substances, together with a great impoverishment of neural nutrition [357,358,359]. The enlargement and widening of PVS, lead a consequent leakage of fluid and plasma cells due to an obstructive process maintained by catabolites, proteins, and cell debris [35], together with the disruption of the BBB, which might eventually potentiate the perivascular inflammation, and all of the cascades of the inflammatory/obstructive/stagnation-induced process [360,361,362]. Therefore, many Authors have tried to propose a biochemical profile of SVD, in order to better evaluate and support the radiological progressions of the SVD and its clinical monitoring [320,363,364,365,366] Low et al. [367] systematically reviewed the existing literature on the associations between markers of inflammation and SVD in cohorts of older people with good health, cerebrovascular disease, or cognitive impairment. 

Their review is focused on systemic inflammation markers (e.g., C-reactive protein, interleukin-6, fibrinogen) or vascular inflammation/endothelial dysfunction (e.g., homocysteine, von Willebrand factor, Lp-PLA2). Their work [367] evidenced a strong association between the SVD and markers of vascular inflammation; nevertheless, the most substantial results have been found in stroke patients. Conversely, cross-sectional findings on systemic inflammation were diversified, with a tendency of an intimate relationship between elevated levels of systemic inflammatory markers and SVD severity and progression [367]. Further studies should be dedicated to the topic, refining clinical criteria of choice and the historical progression of SVD. Their results will help to detect a potential therapeutic target on the specific functional causative processor on some of its markers (homocysteine, i.e.).

## 9. Conclusions

At the end of this review, there are some reliable conclusions and many points that should be addressed by further studies. 

The constructive and accepted aspects are: Small vessel disease and related dementia are two unambiguous nosographic entities that deserve attention for their relevance and impact in clinical practice.The arterioles are the targets of small vessel disease, with degeneration of the smooth muscle layer and replacement by hyaline fibrosis, leading to a subtotal luminal occlusion.To be added, the more intriguing emerging aspects are those concerning the altered endothelium activation, astrocytes modifications, microglia activation, pericytes, and BBB disruption.SVD is related to altered neurovascular coupling.The SVD is an ongoing process, which begins with altered microvessels and pial arteries and ends in a subcortical dementia; CBF regional selective decrease seems to be one of the critical factors for the progression from small vessel disease to small vessel disease-related dementia.Altered response to inflammation, oxidative stress are crucial aspects of possible irreversible post-transcriptional modifications: these processes are auto-expanding features of irreversible changes of a primarily aging-related benign process.Neuromodulators, i.e., Acethlycholine, GABA, and endothelium-acting molecules, such as NO, VEGF, ICAM, prostanoids interact, and are deeply involved in the perpetuation of the ongoing pathological cascade of events.

The most relevant still under debate-aspects are:The modality of changing from typical aging small vessel disease towards dementia.The impact of aging and the age-gene interactions in arteriolosclerosis, endothelium dysfunction, pericytes alterations, and astrocytes modifications, which are caused, promoted, or potentiated by hypoperfusion and metabolic disruption.The altered neurovascular coupling is a secondary or a causative (primary) defect in SVD.The potential clinical rescue from SVD, knowing different times of its development, probably by acting precociously, simultaneously via various strategies (anti-inflammatory, anti-oxidant, anti-thrombotic, and perhaps via nutraceutical promotions [368], acting as co-enzymatic promoters.

Further and dedicated studies should be done to debate, understand, and answer these questions.

## Figures and Tables

**Figure 1 ijms-21-01095-f001:**
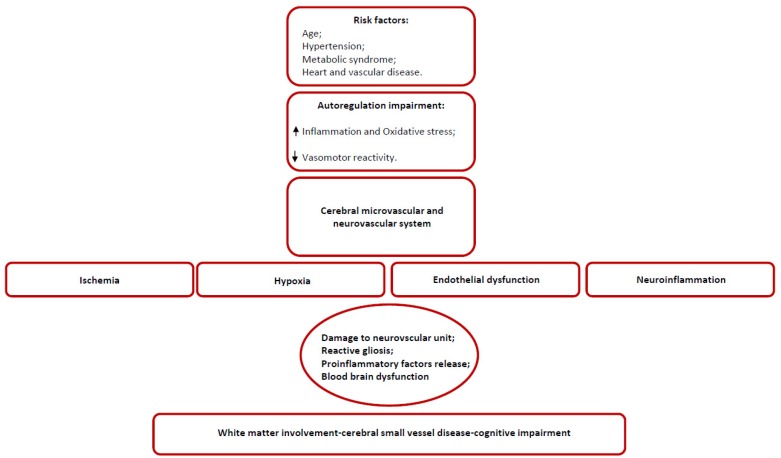
A synopsis of the possible superimposing factors conditioning the progression of SVD.

**Figure 2 ijms-21-01095-f002:**
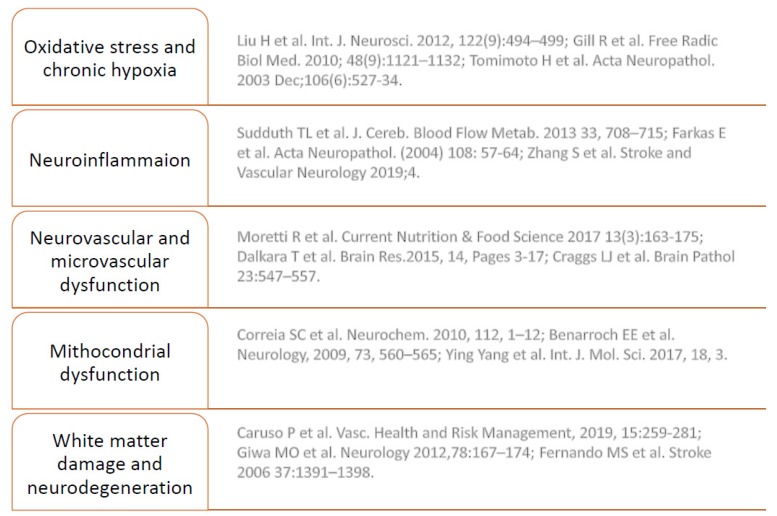
Confirmed pathological processes underlying SVD.

**Figure 3 ijms-21-01095-f003:**
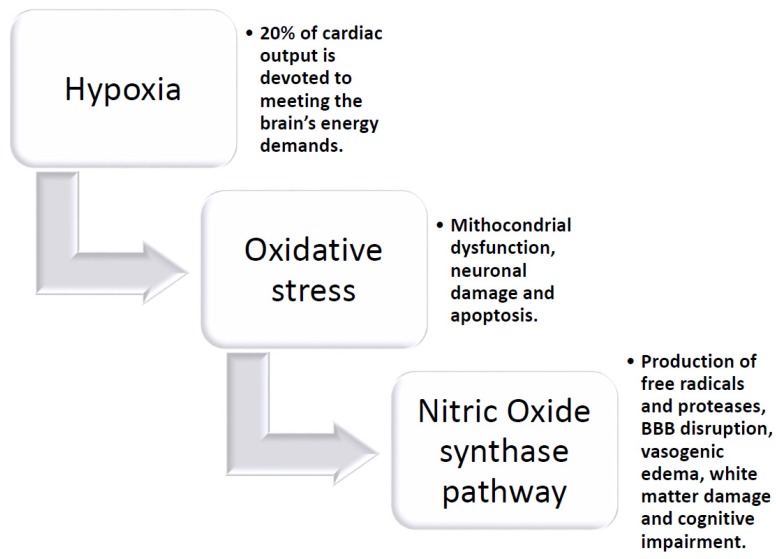
The pathological circuit of chronic hypoxia damage in the brain.

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
