# Peer review of "Small Vessel Disease-Related Dementia: An Invalid Neurovascular Coupling?"

_ijms, 2020, doi:10.3390/ijms21031095_

Round 1

Reviewer 1 Report

In a comprehensive review, the authors have described possible mechanisms which contribute to the pathology of SVD and sub-cortical dementia. Overall, it is generally well written and organised. I only have minor comments:

There should be a paragraph dedicated to the inflammatory response in SVD and dementia. The involvement of peripheral immune cells should be included.  There is emerging evidence that alterations in the gut microbiota can play an important role in the development of SVD and cognitive impairment. This should be discussed. Other risk factors for SVD such as obesity should be mentioned.  It would be beneficial to add diagrams that illustrate concepts and potential mechanisms. There are some inconsistencies in the use of abbreviations. Eg. Cerebral small vessel disease is referred to as cerebral SVD and CSVD.  Some of the paragraphs (eg. Second paragraph under sub heading: Anatomical and structural weaknesses in small vessel disease) are very long which can make it difficult to read. It would be helpful if there were more subheadings or line breaks in the manuscript. The manuscript would benefit from more thorough proof reading. There are quite a few typos. Eg. Conclusion point 2:The arterioles are the target of small vessel disease and SVD.

Author Response

Dear Sir, thank you for your kind reply and precise revision.

We have added a list of abbreviations to be added at the beginning of the work and corrected the list inside the text.

We have corrected the text, amending the typos; hopefully, it would fit well.

We have amended conclusions point 2.

We have added three figures/diagrams to illustrate the concepts and potential mechanisms of SVD.

Finally, we have added a new paragraph, concerning the inflammatory response and the peripheral immune response, including gut microbiota and potential peripheral biomarkers in SVD.

Thank you again.

Reviewer 2 Report

In this manuscript the authors provide a thourough and very extensive review of the current knowledge on the molecular mechanisms, cellular defects and anatomical changes involved in small vessel disease-related dementia.

The manuscript is quite well organized. The authors first provide a definition of small vessel disease, then they introduce vascular-related dementia and finally provide an in-depth review of current knowledge of the role played by endothelial cells, neuroinflammation (astrocytes and microglia activation), BBB dysruption and oxydative stress in the disease.

The chapter on the role of ROS and possible involvement of NADPH-oxidase is up-to date and provide interesting hints on new potential therapeutic approaches.

However, overall the manuscript requires some editing to improve readability:

The authors use several abbreviations in the manuscript. Many of these abbreviations are not explained in the text or are misleading. The authors should explain each abbreviation in full the first time they are encountered in the text. I also suggest to add a paragraph with a list of all the abbreviations used and verify consistency across the text. e.g.: line 28. Cerebral small vessel disease is abbreviated SVD...then at line 36 the authors write CSVD; line 13 small vessel disease is abbreviated SVD. Then the authors also add sVAD without explaining.... The conclusion should be revised to make each statement more clear. e.g. line 678 (In statement 3 it's not clear to me what is the message that the authors want to underline); line 680. I am not sure that statement 4 fits in this list. The authors are summarizing concluding remarks on what is accepted and evident from scientific evidence. I suggest to move statement 4 to a "what to do next" section. I believe that the impact and usefulness of the manuscript would benefit if the authors would add a figure or a table listing the most relevant evidences on the molecular mechanisms of vascular-related dementia (with references).

Author Response

Thank You Sir, for your work and for your encouraging words:

We have added a list of abbreviations, to be inserted at the beginning of the work, and corrected them inside the work.

Line 678 and 680 have been amended; moreover,  point 4 has been moved in the following section.

We have included three figures, inside the text, with references, to better explain the text.

Once more, thank you for your work

Reviewer 3 Report

The paper by Moretti, R. is interesting. and contributes scientifically to the Dementia field. I have few comments to improve the manuscript:

It would look more organized if the authors can able to separate the several sections of the manuscript according to the 'Introduction', 'Details', 'Conclusion', 'Discussion', and 'Acknowledgement' like a conventional manuscript. The paper would also look more impressive when the authors add a figure and a table to describe the details of the paper. The major factors related to the SVD can be tabulated in the form of a table. One of the factors like 'Chronic hypoxia and brain response' can be represented by an image. This will attract the readers from the other fields as well.

Author Response

Dear Sir, 

thank you for your words and appreciation.

We have organized the manuscript in Introduction, Detail (with different sub-heading), Discussion and Conclusion. 

We have inserted three figures/diagrams, inside the work, one of them concerning the major risk factors related to SVD and a diagram concerning Chronic Hypoxia. 

Once again, thank you for your work